

# The effect of rehabilitation time on functional recovery after arthroscopic rotator cuff repair: a systematic review and meta-analysis

Yang Chen[1,*], Hui Meng[2,*], Yuan Li[1], Hui Zong[1], Hongna Yu[1], HaiBin Liu[3], Shi Lv[4] and Liang Huai[3]

[1] Department of Rehabilitation, Taian Maternal and Child Health Hospital, Taian, Shandong, China
[2] Department of Joint and Sports Medicine, Second Affiliated Hospital of Shandong First Medical University, Taian, Shandong, China
[3] School of Sports Medicine and Rehabilitation, Shandong First Medical University, Taian, Shandong, China
[4] Department of Rehabilitation, Second Affiliated Hospital of Shandong First Medical University, Taian, Shandong, China
[*] These authors contributed equally to this work.

Corresponding authors
Shi Lv, ls15666080332@163.com
Liang Huai, hlniren@163.com

## ABSTRACT

**Objective.** We compared the effects of early and delayed rehabilitation on the function of patients after rotator cuff repair by meta-analysis to find effective interventions to promote the recovery of shoulder function.

**Methods.** This meta-analysis was registered in PROSPERO (CRD42023466122). We manually searched the randomized controlled trials (RCTs) in the Cochrane Library, Pubmed, Cochrane Library, EMBASE, the China National Knowledge Infrastructure (CNKI), the China VIP Database (VIP), and the Wanfang Database to evaluate the effect of early and delayed rehabilitation after arthroscopic shoulder cuff surgery on the recovery of shoulder joint function. Review Manager 5.3 software was used to analyze the extracted data. Then, the PEDro scale was employed to appraise the methodological quality of the included research.

**Results.** This research comprised nine RCTs and 830 patients with rotator cuff injuries. According to the findings of the meta-analysis, there was no discernible difference between the early rehabilitation group and the delayed rehabilitation group at six and twelve months after the surgery in terms of the VAS score, SST score, follow-up rotator cuff healing rate, and the rotator cuff retear rate at the final follow-up. There was no difference in the ASES score between the early and delayed rehabilitation groups six months after the operation. However, although the ASES score in the early rehabilitation group differed significantly from that in the delayed rehabilitation group twelve months after the operation, according to the analysis of the minimal clinically important difference (MCID), the results have no clinical significance.

**Conclusions.** The improvement in shoulder function following arthroscopic rotator cuff surgery does not differ clinically between early and delayed rehabilitation. When implementing rehabilitation following rotator cuff repair, it is essential to consider the paradoxes surrounding shoulder range of motion and tendon anatomic healing. A program that allows for flexible progression based on the patient's ability to meet predetermined clinical goals or criteria may be a better option.

## INTRODUCTION

One of the most common causes of shoulder pain and dysfunction is rotator cuff injuries (RCI), which account for roughly 50% of shoulder disorders (*Mazzocca et al., 2017*). As people age, the incidence rises, with rotator cuff injuries affecting approximately 25.6% of those over 60 and up to 45.8% of those over 70 (*Luo et al., 2022*). Rotator cuff repair (RCR) is the suggested course of action for people for whom conservative therapy is not working (*Osborne et al., 2016*). Compared to other surgical modalities, RCR had comparable operative results and a reduced risk of complications. A randomized controlled trial was designed to compare the clinical outcomes of minor open repair *versus* total arthroscopic repair of rotator cuff tears. The clinical outcomes were estimated using the Constant-Murley scores, angular changes in the range of motion, questionnaires from the Disabilities of the Arm, Shoulder, and Hand, and Visual Analog Scale. The findings demonstrated that Arthroscopic repair is related to less pain, lower DASH score, and higher CMS score, arthroscopic surgery had a better prognosis at the short-term follow-up (*Liu et al., 2017*). Two hundred seventy-three patients from 19 teaching and general hospitals in the United Kingdom were enrolled in a randomized controlled trial with the Oxford Shoulder Score at two years postoperatively as the primary outcome indicator to compare the efficacy of arthroscopic repair with open surgery after degenerative rotator cuff tears. The study demonstrated that arthroscopic repair produced better recovery results (*Carr et al., 2017*). Although arthroscopic surgery lessens acute damage to the overall shoulder structure, postoperative stiffness may still happen and negatively impact the patient's functional system and quality of life (*Tauro, 2006*). For this reason, postoperative rehabilitation regimens are essential to getting the best results. However, the best time to recuperate after arthroscopic treatment is still being discussed. Early mobilization proponents stress that this significantly lowers postoperative shoulder stiffness, a frequent consequence after RCR (*van der Meijden et al., 2012*). On the other hand, other studies disagree, contending that ankylosis may not increase with longer braking durations (*Parsons et al., 2010*). Delaying exercise improves rotator cuff shape, composition, and biomechanical qualities while lowering stress at the healing site and lowering the chance of recurrent soft tissue re-dissection, according to animal studies (*Thomopoulos, Williams & Soslowsky, 2003*). The goals of optimal rehabilitation programs are to restore normal shoulder function, allow tendons to recover, and prevent re-injury (*Longo et al., 2020*). Options for early and delayed recovery have been made to accomplish this aim. After rotator cuff repair, immobilization is advised for 6 to 8 weeks, according to the deferred rehabilitation treatment plan. The information from animal research demonstrating that the tendon healing process takes 4 to 16 weeks (*Gimbel et al., 2007*; *Peltz et al., 2010*) is the basis for the theoretical rationale behind this strategy. Although early rehabilitation programs that permit activity to start on the first postoperative day theoretically minimize postoperative stiffness and muscle

atrophy, some research indicates that in 20% to 90% of instances, they increase the chance of tendon re-injury (*Kim et al., 2012*; *Lv et al., 2022*). Clinical trials with varying designs yield inconsistent findings on the best rehabilitation duration and cannot generate substantial proof about the best rehabilitation regimen following RCR. Many RCTs have been carried out recently to examine if an early rehabilitation program is superior to a delayed rehabilitation program in helping shoulder function recover. To provide patients with an efficient rehabilitation program following arthroscopic rotator cuff repair by retrieving and quantifying the results of early and delayed rehabilitation following relevant rotator cuff injuries through randomized controlled trials, this study aimed to find effective interventions to promote functional recovery of the shoulder joint.

## MATERIALS AND METHODS

### Inclusion and exclusion criteria

#### Inclusion criteria

Study inclusion criteria should follow the PICOS principles. P (Population): Patients aged 18 years and older who underwent arthroscopic repair of at least one total rotator cuff tendon tear. I (Intervention): Early physical rehabilitation (passive activity begun within two weeks after surgery) within two weeks after surgery. C (Comparison): With the primary purpose of promoting tendon healing, delayed rehabilitation activities were started 4–6 weeks after surgery, and delayed rehabilitation started 4–6 weeks after surgery. O (Outcome): We only retrieved the data based on the following scales, even if the researcher employed other functional scales. 1. Visual Analog Score (VAS). 2. American Shoulder and Elbow Surgeons (ASES) scores (*Tie et al., 2021*). 3. Simple shoulder test (SST) scores (*Kanto et al., 2021*). 4. Shoulder joint range of motion (ROM). 5. Follow-up rotator cuff healing rate. 6. The incidence of rotator cuff re-tear in the last follow-up. S (Study Design): Only RCTs comparing delayed and early rehabilitation strategies are included.

#### Exclusion criteria

Review, non-English/Chinese studies, Research on the inability to get clear outcome indicators, Partial case data, and duplicate studies.

### Studies screening and quality evaluation

This meta-analysis was registered in PROSPERO (CRD42023466122). Two researchers (Yang Chen and Yuan Li) repeatedly skimmed titles and abstracts to identify all pertinent papers, comparing the entire article to predetermined standards. Any disagreements were discussed with a third investigator (Huai Liang). Cochrane Library, Pubmed, Cochrane Library, EMBASE, China National Knowledge Infrastructure (CNKI), China VIP Database (VIP), and Wanfang Database are among the databases used for retrieval. The database may only be retrieved between 1 September 2023 and the date of establishment. English searches included the following keywords: Arthroscopy, rotator cuff, rehabilitation, early/aggressive/accelerate/delay/conservative/traditional, rehabilitation/exercise, randomized controlled trial. Chinese search keywords: arthroscopy, rotator cuff injury, rotator cuff repair, early rehabilitation, conservative rehabilitation, and
randomized controlled trial. Two investigators independently determined whether the studies met the inclusion criteria in study screening, in cases where two investigators could not agree, they conferred with the third to decide whether or not to accept. The risk of study bias was evaluated from seven items, including random sequence generation, allocation concealment, blinding of investigators and subjects, blinded evaluation of student outcomes, completeness of outcome data, selective reporting of study results, and other sources of bias. Two investigators strictly adhered to the risk assessment tool for discrimination provided in the Cochrane Handbook (*Higgins et al., 2011*). According to the fulfillment of the criteria, "high risk of bias," "low risk of bias," and "unclear" were given. If a study met all of the above criteria, the study's quality grade was set to "A," indicating a low likelihood of bias, if it met some of the requirements, the study's quality grade was set to "B," meaning a moderate probability of discrimination. If it did not meet the criteria, the study's quality grade was set to "C," and C-grade studies would be excluded from the study.

## Publication bias analysis

When there are less than ten included studies for a meta-analysis of outcome metrics, it is generally not suggested to utilize funnel plots for publication bias analysis, only subjective publication bias analysis was carried out (*Chang et al., 2015b*). 1. Because there is a higher chance of publication bias due to the limited sample size of RCTs in this study (*Egger et al., 1997*). 2. Although this study only selected high-quality research to increase the strength of the research results, there are still some limitations. The conservative protocol used in this included RCTs, which was the same, whereas the early rehabilitation group used a protocol with different exercise times and sessions per week, in some studies, the first postoperative activity was by pendulum and CPM (*Arndt et al., 2012*). In others, the first postoperative activity was fully assisted by a therapist (*Cuff & Pupello, 2012*), and given this extensive heterogeneity, it was impossible to perform a subgroup analysis. 3. There was heterogeneity in the extent of patients' rotator cuff injuries, surgical approach, and rehabilitation design among RCTs. 4. Inclusion of only English- and Chinese-language RCTs may be subject to publication bias.

## Quality assessment

The PEDro scale was used to assess the research's methodological quality. This collection consists of ten components, each of which received a present (Y) or absent (N) rating (Table S1). Ten points in total were obtained by adding them together. The research was categorized as low methodological studies (<6 points) and high methodological studies (≥6 points) based on the Pedro score (*Maher et al., 2003*). Three authors assessed the potential for bias in the papers included in the analysis using the PEDro scale. In instances of disagreement, a third researcher was responsible for making the ultimate determination.

## Statistical analysis

The retrieved effect indicators were subjected to a meta-analysis using the RevMan5.3 program. Risk ratio (RR) and 95% confidence interval (CI) are used to describe dichotomous variables, and mean difference (MD) and 95% CI are used to represent

continuous variables. When $P > 0.1$, the heterogeneity is $I^2 \leq 50\%$. Fixed model effect analysis is utilized. A random effect model is conducted when $P < 0.1$ and $I^2 > 50\%$. Statistics were deemed significant if $P < 0.05$. The minimal clinically important difference (MCID) was used to examine the findings from statistically diverse assessment scales. Calculation of MCID values according to the anchor method. MCID = $M_1$–$M_2$ ($M_1$ and $M_2$ are the standard mean values before and after treatment respectively). The score difference was clinically significant if it was larger than the MCID. The MCID is ∼6.4 for the ASES. And have not been defined for the SST (*Roy, MacDermid & Woodhouse, 2009*).

## RESULTS

### Study search results

Four hundred eighty-six different types of studies were retrieved using the predetermined retrieval approach. Excluded were duplicates, non-randomized controlled trials, inconsistent intervention measures, inconsistent outcome indicators, and data that could not be recovered. These were determined by reading the title and abstract and using the known studies' inclusion and exclusion criteria. A meta-analysis of nine studies was conducted (*Arndt et al., 2012*; *Cuff & Pupello, 2012*; *Keener et al., 2014*; *Kim et al., 2012*; *Lee, Cho & Rhee, 2012*; *Mazzocca et al., 2017*; *Sheps et al., 2015*; *Rui et al., 2018*; *Chenglong, Hua & Gang, 2015*). This research comprised 830 patients with rotator cuff injuries, 426 in the experimental group and 404 in the control group. All studies compared patients' general information, such as age, gender, and course of disease, and the difference was not statistically significant ($P > 0.05$). Figure 1 depicts the process of screening studies. The essential characteristics of the nine studies included in the study are shown in Table 1. The Cochrane Handbook was used to evaluate the potential for bias in the included papers. The findings demonstrated that all documents were categorized according to the randomized control principle, three studies using the randomized table of numbers method for random assignment, two studies using computerized randomization, and the remaining four studies did not specify. Blinding was not used except for strict double blinding in *Sheps et al. (2015)* And investigator single blinding in *Mazzocca et al. (2017)* (Table 2). Selective publication and other biases were not found in all the studies, and the results are shown in Figs. 2 and 3.

Table S1 displays the methodological quality assessment. The range of PEDro ratings was between 7 and 9. Each analysis included in the survey was rated "high methodological quality."

### Meta-analysis results and evidence evaluation quality results
#### *Pain score*
The meta-analysis indicates that pain intensity evaluated with VAS does not present statistically significant differences between the groups six months after rotator cuff surgery ($n = 6$; MD = 0.03; 95% CI [−0.20–0.26], $P = 0.80$) (Fig. 4), nor at twelve months ($n = $; MD = 0.16; 95% CI [−0.33–0.65], $P = 0.52$) (Fig. 5).

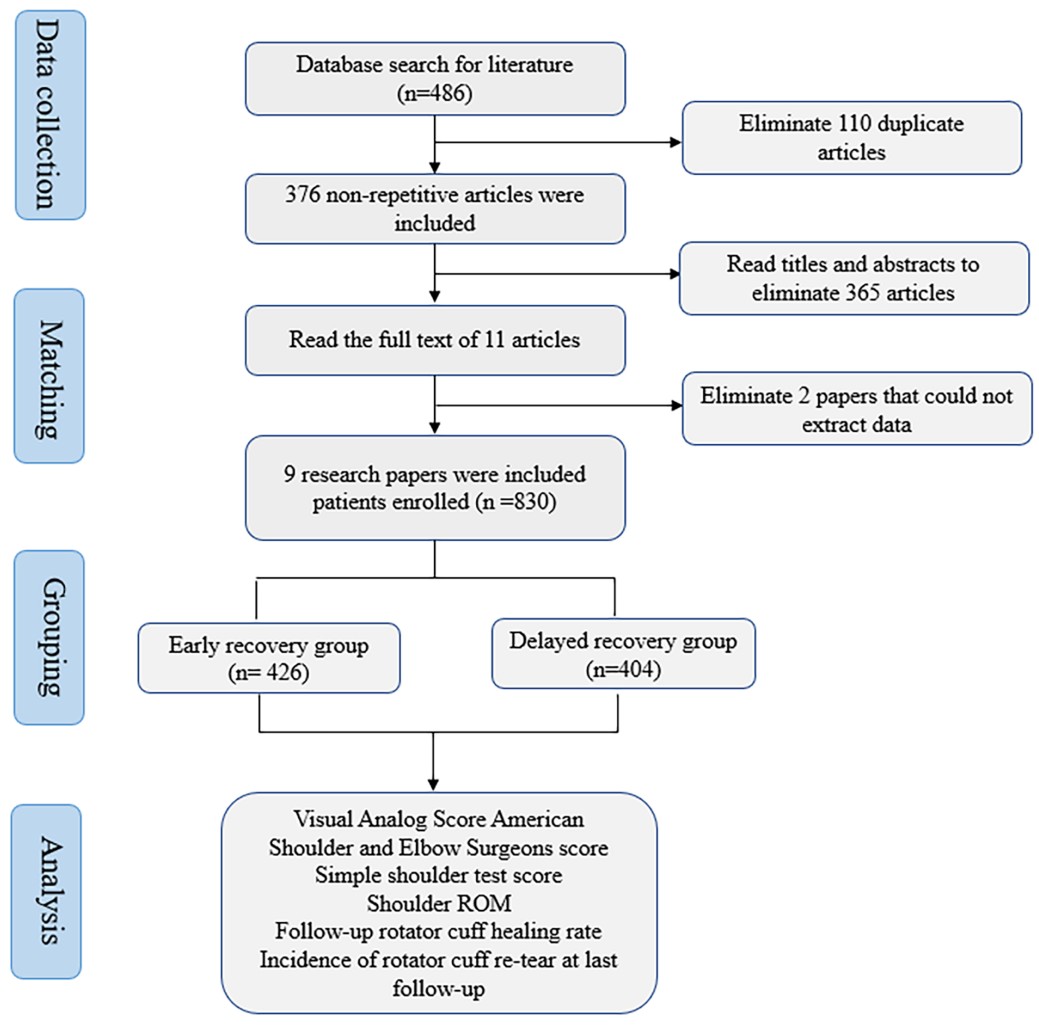

**Figure 1** The flow chart of literature screening.

### Shoulder range of motion (ROM)

The meta-analysis indicates that there was a significant statistical difference in the flexion angle of shoulder joint between the two groups 6 months after shoulder cuff surgery ($n = 8$; MD = 4.31, 95% CI [2.16–6.47], $P < 0.0001$) (Fig. 6). Twelve months after shoulder cuff surgery, there was a significant statistical difference in the flexion angle of the shoulder joint between the two groups. the early rehabilitation group can get a better flexion angle ($n = 8$; MD = 1.46, 95% CI [0.15–2.78], $P = 0.03$) (Fig. 7). Six months after shoulder cuff surgery, there was a significant statistical difference between the two groups in Shoulder external rotation angle, the early rehabilitation group can obtain a better external rotation angle. ($N = 8$; MD = 3.22, 95% CI [0.97–5.47], $P = 0.005$) (Fig. 8). Twelve months after shoulder cuff surgery, there was no significant difference in the Shoulder external rotation angle between the two groups ($n = 8$; MD = 1.31, 95% CI [−0.93–3.55], $P = 0.52$) (Fig. 9).

**Table 1** Includes basic features of the literature.

| Author/Year | Number of patients (early/delayed) | Average age (years) | Rehabilitation time after surgery (early/delayed) | Types of rotator cuff injury | Evaluation indicators | Follow-up time (months) |
|---|---|---|---|---|---|---|
| Arndt et al. (2012) | 92<br>49/43 | 55.3 | 1day<br>Six weeks | Supraspinatus full-thickness tear | ④⑤⑥ | 15 |
| Cuff & Pupello (2012) | 68<br>33/35 | 63.2 | 2days<br>6weeks | Full-thickness crescent tear of supraspinatus | ②③④⑤⑥ | 12 |
| Kim et al. (2012) | 105<br>56/49 | 60.0 | 1day<br>4-5weeks | Minor to medium-full thickness tear | ①②③④⑤⑥ | 12 |
| Lee, Cho & Rhee (2012) | 64<br>30/34 | 54.8 | 1day<br>6weeks | Medium-to-large full-thickness tear | ①④⑤⑥ | 12 |
| Keener et al. (2014) | 114<br>61/53 | 55.3 | 1day<br>6weeks | Minor to medium-full thickness tear | ①②③④⑤⑥ | 24 |
| Sheps et al. (2015) | 189<br>97/92 | 55.1 | 1day<br>6weeks | Full thickness tear | ①④⑥ | 24 |
| Mazzocca et al. (2017) | 73<br>37/36 | 54.5 | 2-3days<br>4weeks | Supraspinatus full-thickness tear | ①②③④⑤ | 24 |
| Huang, Wu & Cheng (2015) | 65<br>33/32 | 55.1 | 2days<br>6weeks | Medium-to-large full-thickness tear | ①③④⑤⑥ | 12 |
| Guo, Wang & Dong (2019) | 60<br>30/30 | 57.1 | 2days<br>4-7weeks | Full thickness tear | ①③ | 12 |

Notes.
①, Visual Analog Score (VAS); ②, American Shoulder and Elbow Surgeons (ASES) score; ③, Simple shoulder test (SST) score; ④, Shoulder range of motion (ROM) ); ⑤, rotator cuff healing rate at follow-up; ⑥, rotator cuff re-tear rate at last follow-up.

**Table 2** Bias-dependent risk assessment of included studies.

| Author/Year | Random sequence generation | Allocation concealment | Blinding of participants and personnel | Blinding of outcome assessment | Incomplete outcome data | Selective reporting | Other bias |
|---|---|---|---|---|---|---|---|
| Arndt et al. (2012) | Unclear | Unclear | N | Y | 5 Lost | N | Unclear |
| Cuff & Pupello (2012) | Random number table | Airtight envelope | N | Y | Complete | N | Unclear |
| Kim et al. (2012) | Random number table | Unclear | N | Y | 12Lost | N | Unclear |
| Lee, Cho & Rhee (2012) | Unclear | Unclear | N | Y | 6Lost | N | Unclear |
| Keener et al. (2014) | Computer random | Airtight envelope | N | Y | 10Lost | N | Unclear |
| Sheps et al. (2015) | Random number table | Airtight envelope | Y | Y | Complete | N | Unclear |
| Mazzocca et al. (2017) | Computer random | Electronic document | Y | Y | 11Lost | N | Unclear |
| Huang, Wu & Cheng (2015) | Unclear | Unclear | N | Y | 2Lost | N | Unclear |
| Guo, Wang & Dong (2019) | Unclear | Unclear | N | Y | Complete | N | Unclear |

### Shoulder joint function score

The meta-analysis indicates that there was no statistically significant difference in ASES score between the two groups 6 months after shoulder cuff surgery ($n = 3$; MD = 0.33; 95% CI [$-6.50$–7.17], $P = 0.92$) (Fig. 10). Twelve months after shoulder cuff surgery, there was a statistical difference in ASES score between the two groups ($n = 4$; MD $= -1.56$, 95% CI [$-2.66$ to $-0.46$], $P = 0.006$) (Fig. 11). Six months after shoulder cuff surgery, there was no statistically significant difference in SST scores between the two groups ($n = 5$; MD = 0.40, 95% CI [0.16–0.97], $P = 0.16$) (Fig. 12). In addition, there was no statistically

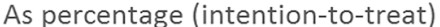
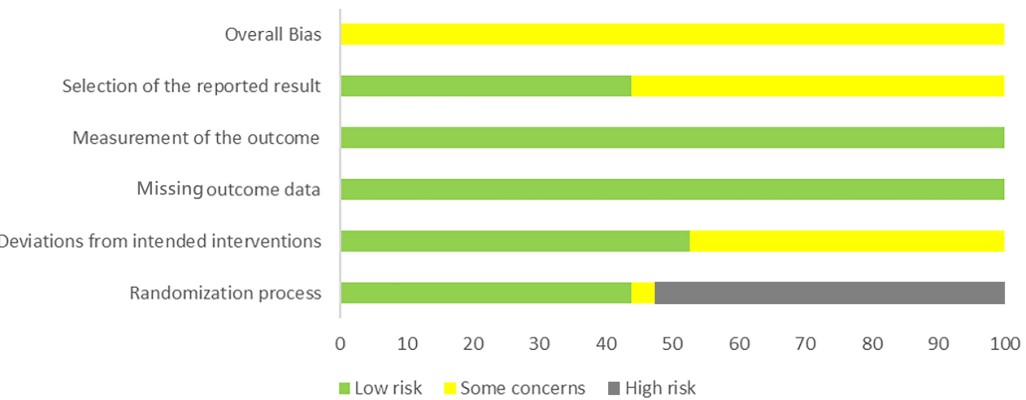

**Figure 2** The risk assessment of biased dependence of the included studies.

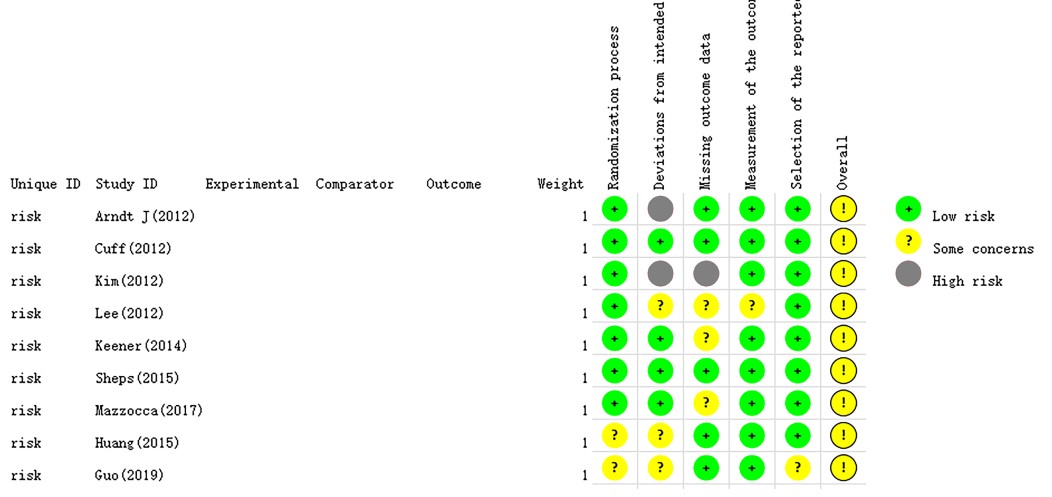

**Figure 3** Summary graph of risk of bias of included studies.

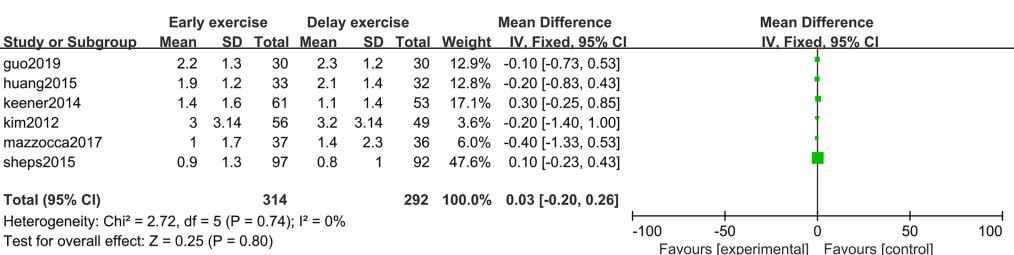

**Figure 4** VAS score six months after surgery.

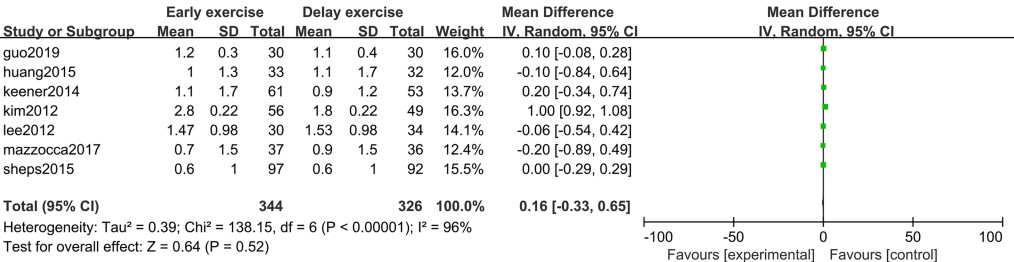

**Figure 5** VAS score 12 months after surgery.

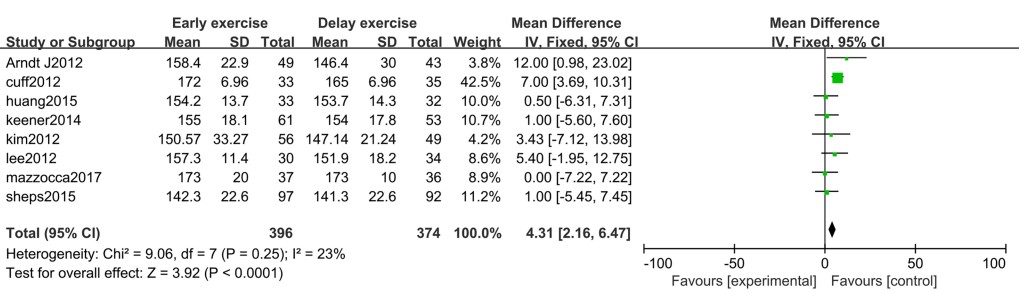

**Figure 6** Range of motion in forward flexion six months after surgery.

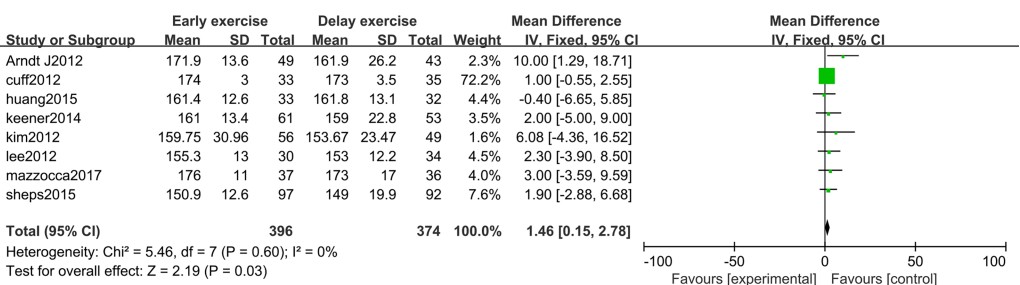

**Figure 7** Range of motion in forward flexion 12 months after surgery.

significant difference in SST scores between the two groups 12 months after shoulder cuff surgery ($n = 6$; MD = 0.28, 95% CI [−0.05–0.60], $P = 0.09$) (Fig. 13).

### Postoperative follow-up rotator cuff healing rate and re-tear rate

The meta-analysis indicates that there was no statistically significant difference in the healing rate of rotator cuff repair between the two groups 12 months after rotator cuff surgery ($n = 7$; RR = 0.80, 95% CI [0.52–1.24], $P = 0.32$) (Fig. 14). After 12 months of rotator cuff surgery, there was no statistically significant difference in rotator cuff re-tearing rate between the two groups ($n = 7$; RR = 0.63, 95% CI [0.39–1.02], $P = 0.06$) (Fig. 15).

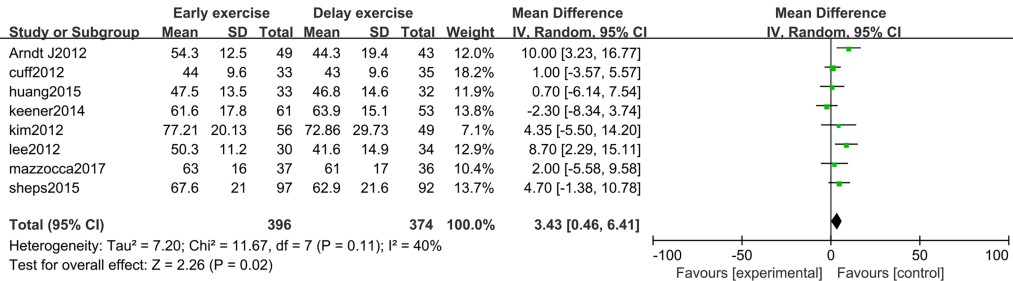

**Figure 8** The range of motion of shoulder external rotation six months after surgery.

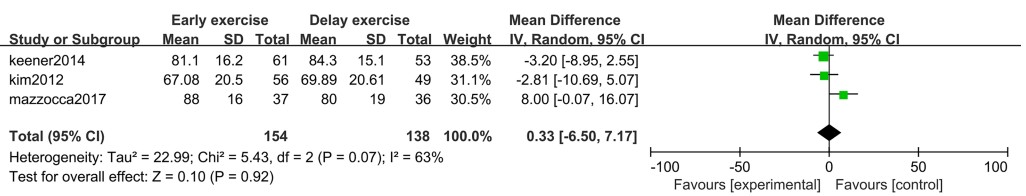

**Figure 9** The range of motion of shoulder external rotation 12 months after surgery.

**Figure 10** ASES score six months after surgery.

## DISCUSSION

The advantages and disadvantages of early *versus* delayed rehabilitation following arthroscopic rotator cuff repair have been extensively studied (*Dickinson et al., 2017*; *Mazuquin et al., 2018*). However, based on the available evidence, the American Academy of Orthopaedic Surgeons (AAOS) could not definitively conclude the optimal rehabilitation time and the program (*Tashjian, 2011*). Proponents of early rehabilitation argue that early rehabilitation reduces shoulder stiffness, muscle atrophy, and other complications (*van der Meijden et al., 2012*). In contrast, delayed rehabilitation advocates argue that delayed motion in postoperative patients is more consistent with the physiological properties of tendon-bone healing (*Parsons et al., 2010*). Therefore, this research performed a meta-analysis of published RCTs domestically and internationally to gather the best available data about the ideal rehabilitation regimen after arthroscopic rotator cuff surgery. The
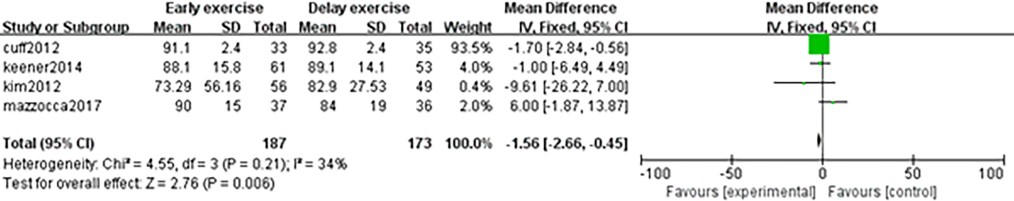

**Figure 11  ASES score 12 months after surgery.**

| Study or Subgroup | Early exercise Mean | SD | Total | Delay exercise Mean | SD | Total | Weight | Mean Difference IV, Fixed, 95% CI |
|---|---|---|---|---|---|---|---|---|
| guo2019 | 8.8 | 2.9 | 30 | 8.9 | 3 | 30 | 14.4% | -0.10 [-1.59, 1.39] |
| huang2015 | 9.3 | 2.7 | 33 | 9.1 | 2.6 | 32 | 19.3% | 0.20 [-1.09, 1.49] |
| keener2014 | 9.1 | 2.7 | 61 | 9.3 | 2.9 | 53 | 30.0% | -0.20 [-1.23, 0.83] |
| kim2012 | 7.81 | 3.24 | 56 | 6.7 | 3.64 | 49 | 18.2% | 1.11 [-0.22, 2.44] |
| mazzocca2017 | 10 | 2.4 | 37 | 8.7 | 3.3 | 36 | 18.2% | 1.30 [-0.03, 2.63] |
| **Total (95% CI)** | | | 217 | | | 200 | 100.0% | 0.40 [-0.16, 0.97] |

Heterogeneity: Chi² = 4.69, df = 4 (P = 0.32); I² = 15%
Test for overall effect: Z = 1.40 (P = 0.16)

**Figure 12  SST score six months after surgery.**

| Study or Subgroup | Early exercise Mean | SD | Total | Delay exercise Mean | SD | Total | Weight | Mean Difference IV, Fixed, 95% CI |
|---|---|---|---|---|---|---|---|---|
| cuff2012 | 5.5 | 0.93 | 33 | 5.1 | 0.93 | 35 | 53.1% | 0.40 [-0.04, 0.84] |
| guo2019 | 10.8 | 1.7 | 30 | 11 | 2 | 30 | 11.8% | -0.20 [-1.14, 0.74] |
| huang2015 | 10.8 | 1.9 | 33 | 10.7 | 2 | 32 | 11.5% | 0.10 [-0.85, 1.05] |
| keener2014 | 10.8 | 1.8 | 61 | 10.6 | 2.5 | 53 | 15.8% | 0.20 [-0.61, 1.01] |
| kim2012 | 9 | 5.45 | 56 | 9 | 4.7 | 49 | 2.8% | 0.00 [-1.94, 1.94] |
| mazzocca2017 | 10.2 | 2.6 | 37 | 9.3 | 3.6 | 36 | 5.0% | 0.90 [-0.54, 2.34] |
| **Total (95% CI)** | | | 250 | | | 235 | 100.0% | 0.28 [-0.05, 0.60] |

Heterogeneity: Chi² = 2.25, df = 5 (P = 0.81); I² = 0%
Test for overall effect: Z = 1.68 (P = 0.09)

**Figure 13  SST score 12 months after surgery.**

study's findings demonstrated no difference in the patient's VAS scores between the early and delayed rehabilitation groups six months and twelve months following surgery. This suggests there is no difference between the two groups regarding shoulder pain and self-perception in the mid-to-long-term postoperative period and that early rehabilitation does not exacerbate the pain.

Regarding ROM, at the 6-month follow-up, the early rehabilitation group showed more significant improvement in forward flexion and external rotation than the delayed rehabilitation group. At the 12-month follow-up, the long-term efficacy of only bold flexion activity exceeded that of the delayed rehabilitation group. There was no significant difference between the two groups in external rotation mobility. The better recovery of external rotation mobility in the early period was only temporary. Therefore, our results suggest that early rehabilitation may increase postoperative shoulder anterior flexion mobility and that this effect lasts at least 12 months. However, the advantage of early ROM was not present in the external rotation angles. This inconsistent result

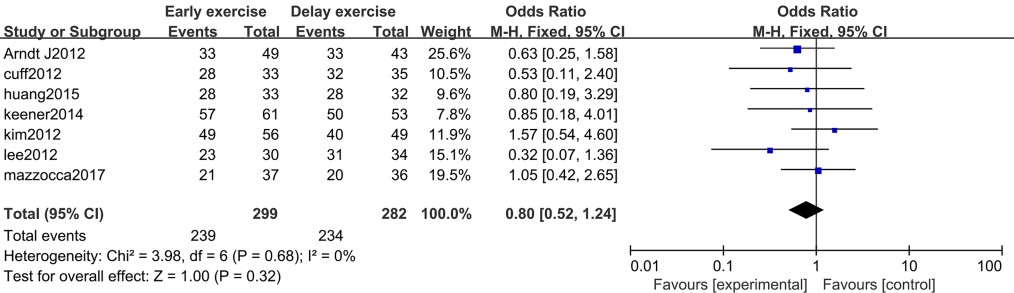

| Study or Subgroup | Early exercise Events | Total | Delay exercise Events | Total | Weight | Odds Ratio M-H, Fixed, 95% CI |
|---|---|---|---|---|---|---|
| Arndt J2012 | 33 | 49 | 33 | 43 | 25.6% | 0.63 [0.25, 1.58] |
| cuff2012 | 28 | 33 | 32 | 35 | 10.5% | 0.53 [0.11, 2.40] |
| huang2015 | 28 | 33 | 28 | 32 | 9.6% | 0.80 [0.19, 3.29] |
| keener2014 | 57 | 61 | 50 | 53 | 7.8% | 0.85 [0.18, 4.01] |
| kim2012 | 49 | 56 | 40 | 49 | 11.9% | 1.57 [0.54, 4.60] |
| lee2012 | 23 | 30 | 31 | 34 | 15.1% | 0.32 [0.07, 1.36] |
| mazzocca2017 | 21 | 37 | 20 | 36 | 19.5% | 1.05 [0.42, 2.65] |
| **Total (95% CI)** | | **299** | | **282** | **100.0%** | **0.80 [0.52, 1.24]** |
| Total events | 239 | | 234 | | | |

Heterogeneity: Chi² = 3.98, df = 6 (P = 0.68); I² = 0%
Test for overall effect: Z = 1.00 (P = 0.32)

**Figure 14** Postoperative rotator cuff healing rate.

| Study or Subgroup | Early exercise Events | Total | Delay exercise Events | Total | Weight | Odds Ratio M-H, Fixed, 95% CI |
|---|---|---|---|---|---|---|
| Arndt J2012 | 16 | 49 | 22 | 43 | 37.4% | 0.46 [0.20, 1.08] |
| cuff2012 | 3 | 33 | 5 | 35 | 10.5% | 0.60 [0.13, 2.74] |
| huang2015 | 4 | 33 | 3 | 32 | 6.3% | 1.33 [0.27, 6.49] |
| keener2014 | 3 | 61 | 6 | 53 | 14.5% | 0.41 [0.10, 1.71] |
| kim2012 | 9 | 56 | 7 | 49 | 14.9% | 1.15 [0.39, 3.36] |
| lee2012 | 3 | 30 | 7 | 34 | 14.0% | 0.43 [0.10, 1.83] |
| sheps2015 | 1 | 97 | 1 | 92 | 2.4% | 0.95 [0.06, 15.38] |
| **Total (95% CI)** | | **359** | | **338** | **100.0%** | **0.63 [0.39, 1.02]** |
| Total events | 39 | | 51 | | | |

Heterogeneity: Chi² = 3.30, df = 6 (P = 0.77); I² = 0%
Test for overall effect: Z = 1.87 (P = 0.06)

**Figure 15** Postoperative rotator cuff re-tear rate.

between grades may be due to initial ROM limitations within the plane of motion of the shoulder. Most procedures are designed to avoid overloading the sutured supraspinatus tendon, and external rotation angles in early ROM protocols are usually limited to 30°. In contrast, shoulder forward flexion is allowed to exceed 90°. However, the point of follow-up assessment for all RCTs included in the study was 12 months after surgery, and we were unable to assess further whether a delayed rehabilitation program would result in a permanent lack of mobility compared with an early rehabilitation program. Because abductor activity was not comprehensively available in some RCTs, its improvement could not be studied.

At six months post-op, there was no significant difference in the ASES and SST scores for joint function improvement between the early rehabilitation group and the delayed rehabilitation group, at the 12-month long-term follow-up, there was no significant difference in the SST score between the two groups. The twelve items on the SST questionnaire are patient-perceived pain and functional recovery measures, compromising the questionnaire's validity and reliability. Although the difference in ASES scores between the two groups was statistically significant, according to *Roy, MacDermid & Woodhouse (2009)*, comparing ASES scores between the two groups in this study 12 months after the operation had no clinical significance. Our research shows no significant difference in the improvement of shoulder joint function between the two schemes.

A further topic of discussion about alternatives for postoperative rehabilitation is the possible effect of exercise on the posterior rotator cuff. According to proponents of delayed restoration, delaying mobilizing the musculature decreases the likelihood of rotator cuff retraction and fretting of tendon-bone healing sites (*Thomopoulos, Williams & Soslowsky, 2003*). Nevertheless, according to our findings, the postoperative rotator cuff healing rate and rotator cuff writer did not significantly vary between the early and delayed rehabilitation groups. This aligns with the findings of *Chan et al. (2014)* and *Chen et al. (2015)*. As a result, methods to lower the rotator cuff tear rate have to be investigated from several angles. Numerous factors—some of which are uncontrollable—such as muscle atrophy and degeneration, more outstanding tears, poor tendon quality (*Shen et al., 2008*), and recurrent injuries—involve the structural failure of tendon healing or repairing tendons. Good clinical and anatomical outcomes can be achieved by fully considering the patient's various preoperative factors and using appropriate surgical techniques and postoperative rehabilitation programs. Controllable factors include those related to the surgical procedure (right surgical approach, identification of tear extent, adequate subacromial decompression, cuff loosening, node preparation, suturing and knotting techniques, anchor placement, and surgeons' experience) and the choice of postoperative rehabilitation programs (*Accousti & Flatow, 2007*; *Millett et al., 2006*).

Despite significant advances and improvements in arthroscopic techniques, postoperative repair failure rates still range from 20% to 90% (*Bartl et al., 2012*; *Khazzam et al., 2012*), as evidenced by recalcitrant joint stiffness, persistent pain, re-tears, and loss of function (*Desmoineaux, 2019*; *Zakko et al., 2019*). Rehabilitation exercises are a critical part of reducing complications to promote functional recovery of the rotator cuff (*Zhang et al., 2013*), and overall, based on the current findings, we do not believe that there are clinically substantial differences between the postoperative exercise regimens included within the RCTs in this study. The implementation of rehabilitation following rotator cuff repair requires additional factors to be considered, weighing the paradoxes of shoulder ROM and anatomical healing of the tendon and a program that provides flexibility of progression based on when the patient can achieve a specific clinical goal or criterion may be more appropriate. Therefore, this study uses a reconstructive rehabilitation model after rotator cuff injury as an example to identify particular guidelines and standards for return to sport.

A patient's functional recovery should involve preoperative rehabilitation. Our clinical experience indicates that preoperative rehabilitation has a variety of impacts. Several investigations have shown the relationship between preoperative and postoperative function. Providing patients with preoperative rehabilitation instruction may help them get the most significant results. A systematic review (*Wang et al., 2016*) assessed the clinical impact of rehabilitation before joint replacement surgery. They searched databases of randomized controlled trials comparing preoperative rehabilitation with no rehabilitation for joint replacement surgery. Postoperative pain and function scores, quality of life, postoperative complications, and adverse events were used as outcome indicators. The study suggests that preoperative rehabilitation may improve joint replacement patients' early postoperative pain and function. A randomized controlled trial (*Brown et al., 2014*) examined whether patients with knee osteoarthritis who underwent guided exercise

before total knee arthroplasty reported higher exercise self-efficacy and higher outcome expectations for exercise than patients who did not. The results suggest that rehabilitation may result in better recovery outcomes. However, it has also been shown that prehabilitation only improves the knee flexion angle in patients with total knee and total hip arthroplasty and has no effect on quality of life and function (*Cavill et al., 2016*). Therapists can ascertain patients' baseline pain, range of motion, and strength by visiting patients before surgery and collaborating with other tests. Patients can also learn about postoperative rehabilitation, including planning for various periods and rehabilitation measures. For patients to share decision-making on their repair and to have early, clear expectations for their postoperative recovery. Within six weeks following arthroscopic surgery, individuals who did not comply with restricted activity had a 152-fold increased relative risk of tendon renter or nonunion (*Ahmad, Haber & Bokor, 2015*). Furthermore, it is essential to take into account several factors that influence tendon recovery, including individual variations (*e.g.*, age, activity level, length of symptoms, size of tear, location of incision, quality of rotator cuff tissue), surgical repair techniques, *etc.* (*Galatz et al., 2005*; *Killian et al., 2012*; *Thomopoulos, Williams & Soslowsky, 2003*). The tendon transfer procedure and subsequent physical therapy strategies used to identify individuals with large rotator cuff (RC) injuries were reported in a scoping review (*Salazar-Méndez et al., 2023*). The study discovered that evidence from existing databases on physical therapy interventions after RC tendon transfer surgery was limited to the number and duration of phases and general characteristics without specifying the type and dose of intervention. Thus, thorough coordination with the surgeon is necessary to determine an acceptable load that will enable the patient to recover and do advanced activities to the fullest extent possible to design an effective rehabilitation plan.

After rotator cuff repair, passive mobility is considered advantageous in the early postoperative phase (*Chang et al., 2015a*). When properly planned, passive range of motion exercises can minimize postoperative loss of motion and protect the repaired rotator cuff. *Salazar-Méndez et al. (2023)* described three phases of physical therapy intervention (early, intermediate, and late) based on the time it took for the tendon to heal. The bleeding phase, for example, required higher care. The inflammatory phase followed (3–7 days) until it reached the proliferative phase (5–25 days) and the remodeling phase (>21 days), which allowed for the gradual integration of high-impact exercises. The healing period of the tendon, which can last anywhere between 4 and 8 weeks, must thus be taken into account in the rehabilitation regimen. Postoperative recovery of Range of action is a priority. We recommend limiting the activity level and dividing it into stages to achieve these two contradictory objectives. The amount of muscle activity, the plane of motion, the precise degree of Range of motion, cyclic loading, and the weight of each particular upper extremity may all impact the tension of the repaired tendon. Exercise recommendations for patients after rotator cuff repair should be based as much as possible on known muscle activity levels, even if these characteristics are crucial since this is the most accurate way to assess rotator cuff tendon tension (*Illyés & Kiss, 2007*; *Michener et al., 2005*; *Reinold et al., 2007*). The amount of pressure applied during rehabilitation training cannot be clinically measured. However, the stress on the rotator cuff may be at least somewhat estimated

using electromyogram (EMG) data (*Thigpen et al., 2016*). Supraspinatus EMG activity levels, which are ≤15% within 1–12 weeks postoperatively, maintained in the range of 16%–29% for 8–16 weeks postoperatively, remained in the field of 30%–49% for 12–20 weeks postoperatively, and reach more than 50% after 20 weeks after surgery (*Gerber et al., 1999*), can be used to classify exercise and predict the pressure acting on the tendon repair.

The primary focus of the regression exercise test is not just the Range of motion but also the restoration of upper body strength. However, once the patient's pain is under control (NPRS < 2 points, out of 10 points) (*Hjermstad et al., 2011*) and enough passive activity is achieved, muscular strength training should continue. Strength training may increase passive and active stress on the rotator cuff repair in addition to reloading, much like phased ROM exercises do. Consequently, we advise starting movement with EMG activity measured at a level of ≤15%. Once the patient can handle active loads, elevate workouts to a 16%–29% EMG activity level (*Thigpen et al., 2016*). Generally, active-assist mobility exercises in this Range should be conducted using gradual movements in postures with the least amount of gravity. In experiments on animals, tendon repairs attained between 25 and 50 percent of their anticipated strength after 12 weeks, and at 15 weeks, tendon bone healing was almost mature (*Gerber et al., 1999*). Additionally, studies show that most individuals may safely raise their EMG activity levels to 30%–49% after 12–20 weeks of strength training (*Sonnabend & Watson, 2002*). After 20 weeks, strength training with an EMG activity level of ≥50% is usually considered safe. However, paying attention to the activities in this process stage is also necessary. According to a study, retails for 2–4 cm injuries often happen during the first six months after surgery (*Iannotti et al., 2013*; *Miller et al., 2011*). Strength training, regardless of its level of intensity, aims to promote tendon regeneration by emphasizing endurance and movement quality while working with relatively light loads. However, it is difficult to quantify how exercise affects tendon biomechanics. The balance between opposing muscle groups may influence muscle performance and injury risk. In clinical settings, muscle performance can be determined by comparing the force ratios of active and antagonistic muscles, which can be used to determine the relative proportion of hostile muscle groups surrounding the joint. The shoulder isokinetic exercise test, which measures the external and internal rotators, has gained reliability in this respect (*Edouard et al., 2011*). The external rotators of the shoulder were assessed for isokinetic muscular strength while seated at 60°/s and 180°/s (45° of shoulder abduction, 30° of flexion, scapula plane). Returning to sports is deemed appropriate if the patient's limb muscle strength (LSI, LSI = limb muscle strength/unaffected limb muscle strength ×100%) is 90% or above (*Davies et al., 2017*). It is also essential to monitor the time it takes to attain peak torque or PT. This displays the time needed for each limb to produce its maximum peak force during the test. The individual being evaluated cannot have the output rapidly, which is necessary for dynamic movement if the peak torque value is the same or comparable across limbs. Still, the test limb takes twice as long to generate that force. Normalizing periarticular active and antagonistic muscle should be considered in addition to LSI, keeping an appropriate ER to IR muscle ratio (66%–75%) may help reduce the likelihood of re-injury (*Wilk, Meister & Andrews, 2002*).

The guiding concepts of exercise rehabilitation centers for distributing the stresses applied to the tissues. Tissue response to loading refers to how the tissue reacts to loads of varying sizes and frequencies. When the load and frequency are balanced, the tissue is loaded to the maximum extent necessary to maximize healing while minimizing adverse effects and preventing tissue re-injury (*Dye, 1996*). Therefore, creating a successful nonsurgical postoperative rehabilitation program requires a thorough grasp of the fundamental biomechanical properties of the tissues. However, we searched the available databases and found few studies on such studies to provide reliable evidence for clinical behavior. Therefore, in future studies, our team will start from this aspect to find the balance between tissue healing ability and loading response and incorporate this information into the patient's rehabilitation program to provide a more scientific and practical rehabilitation program for postoperative orthopedic patients.

### Funding

This work was supported by Key R&D Plan Projects in Shandong Province (2019GSF108203). The funders had no role in study design, data collection and analysis, decision to publish, or preparation of the manuscript.

### Grant Disclosures

The following grant information was disclosed by the authors:
Key R&D Plan Projects in Shandong Province: 2019GSF108203.

### Competing Interests

The authors declare there are no competing interests.

### Author Contributions

- Yang Chen conceived and designed the experiments, authored or reviewed drafts of the article, and approved the final draft.
- Hui Meng conceived and designed the experiments, authored or reviewed drafts of the article, and approved the final draft.
- Yuan Li performed the experiments, prepared figures and/or tables, and approved the final draft.
- Hui Zong analyzed the data, prepared figures and/or tables, and approved the final draft.
- Hongna Yu analyzed the data, prepared figures and/or tables, and approved the final draft.
- HaiBin Liu performed the experiments, prepared figures and/or tables, and approved the final draft.
- Shi Lv performed the experiments, prepared figures and/or tables, and approved the final draft.
- Liang Huai conceived and designed the experiments, authored or reviewed drafts of the article, and approved the final draft.

## Data Availability

The raw data are available in the Supplemental Files.

## Supplemental Information

Supplemental information for this article can be found online at http://dx.doi.org/10.7717/peerj.17395#supplemental-information.

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
