# Peer review of "The effect of rehabilitation time on functional recovery after arthroscopic rotator cuff repair: a systematic review and meta-analysis"

_PeerJ, doi:10.7717/peerj.17395_

## Round 0.1 · original submission · Major Revisions

· Academic Editor

Major Revisions

The manuscript has methodological drawbacks that potentially affect the generalization of the results. However, research has characteristics that make it relevant to the scientific community. Please review and resolve the reviewers' comments.

**Language Note:** The review process has identified that the English language must be improved. PeerJ can provide language editing services - please contact us at [email protected] for pricing (be sure to provide your manuscript number and title). Alternatively, you should make your own arrangements to improve the language quality and provide details in your response letter. – PeerJ Staff

·

Basic reporting

English could be improved.
The references used are relevant. However, there are several statements throughout the manuscript that do not present citations.
The tables and figures are adequate. However, the GRADE table is missing. This analysis indicates having been carried out in the summary but it is not appreciated in the manuscript.

Experimental design

The research is suitable for the journal based on an interesting idea and clinical applicability.
The methodology used could be better specified to highlight a high technical level and replicability.

Validity of the findings

The findings are valid and important in clinical rehabilitation and the conclusions are well directed with respect to the results.

Additional comments

Line 46: “Statistically significant” this is loose without a context. Review the wording

Lines 47-48: “However, there was a statistically significant difference in the ASES score between the early and delayed rehabilitation groups at 12 months after surgery, the minimum clinical” in favor of early or delayed?

Introduction.
Lines 60-61: “Surgical repair is the suggested course of action for people for whom conservative therapy is not working”. I suggest quoting

Lines 61-62: “Compared to other surgical modalities, arthroscopic”. It would be important to mention some examples

Line 67. The abbreviation RCR is not previously defined.

Line 81: “later” before you have used delayed

Lines 83-84: I suggest using the same objective throughout the manuscript since they do not match what you have in the abstract.

Inclusion criteria
Lines 89-91: I suggest replacing “With the primary purpose of promoting rapid recovery of joint function, early rehabilitation was performed within two weeks after surgery” for “early physical rehabilitation (passive activity begun within two weeks after surgery) within two weeks after surgery.”

Lines 91-92: I suggest replacing “Comparison): With the primary purpose of promoting tendon healing, conservative rehabilitation activities were started 4-6 weeks after surgery” for “Conservative rehabilitation started 4-6 weeks after surgery”. I also suggest giving some examples of conservative rehabilitation.

Lines 93-96: “VAS (Visual et al.) 2. Score (14) 3 according to American Shoulder and Elbow Surgeons (ASES). Score (15) 4 on the Simple Shoulder Test (SST). Handle ROM 5. Healing rate of the follow-up rotator cuff: 6. Rate of rotator cuff retear at the last follow-up if the previous follow-up was longer than 12 months;” This needs revision so that there is consistency in the way it has been written. First the scale and in parentheses the abbreviation

Lines 96-97: “Only RCTs comparing conservative and early rehabilitation strategies are included” Its objective is to compare the effects between early and late rehabilitation, not between early and conservative rehabilitation

Lines 97-99: “The delayed rehabilitation group needed at least four weeks of total shoulder immobilization, while early rehabilitation was defined as passive activity begun within two weeks after surgery.” This could be eliminated if they integrate it into the point of intervention and comparison

Exclusion criteria.

Line 101: “unclear rehabilitation programs”, It would be convenient for them to indicate how they are going to determine the studies with little clarity in rehabilitation. lack of dosage, lack of specification in the ranges worked? lack of specification in the movements performed?

Literature screening and quality evaluation

In the abstract they point out that they carry out GRADE but in the methods they do not explain how they do it or the criteria to reduce the level of certainty.

Lines 115-118: This is Risk of Bias 1. Today there is a more updated risk of bias scale, the ROB2, which is carried out using an Excel with integrated macros to determine with algorithms the decision of high risk, some concerns and low risk.


Statistical analysis.

I suggest indicating what the minimum number of studies will be to perform the meta-analysis for each variable. 3 studies minimum? 5 studies minimum?

Results

I suggest integrating a section on characteristics of the studies with some descriptive statistics (% of supraspinatus tears, massive, medium-long; average age or age range found; specific techniques)
I leave you this study as a possible example: doi: 10.1002/pmrj.13089
Then there must be a specific section for risk of bias and you can also use percentages according to the categorizations and domains.

Literaturesearch results

Line 146: I suggest replacing “literature” to “studies” and do it in all the parts where pieces appears.

Line158: I suggest replacing “pieces” to “studies” and do it in all the parts where pieces appears.


3.2.1 pain score.

This could be written in the following way and could be used for the rest of the results report: The meta-analysis indicates that pain intensity evaluated with VAS does not present statistically significant differences between the groups six months after rotator cuff surgery [n=6; MD = 0.03; 95% CI [-0.20; 0.26], P = 0.80] (Figure 4), nor at twelve months [n=7; MD = 0.16; 95% CI [-0.33; 0.65], P = 0.52]] (Figure 5). Both results present a high level of certainty.

3.2.2 Shoulder range of motion (ROM)

Line174: “replacement” I suggest always using the same terminology throughout the manuscript. Surgery, repair, replacement?

Line 175: “The findings indicated that the two groups differed statistically significantly” Here I suggest indicating in favor of which one for those people who do not know how to interpret a forest plot. This should be integrated into all findings in which there are differences.

3.2.4 Postoperative follow-up rotator cuff healing rate and re-tear rate

after how many months?


Discussion:

Lines 204: 28). “Proponents of early rehabilitation argue that early rehabilitation reduces shoulder stiffness, muscle atrophy, and other complications. In contrast, delayed rehabilitation advocates argue that delayed motion in postoperative patients is more consistent with the physiologic properties of tendon bone healing”. These assertions are extremely important, which is why they must be cited.

Lines 261-262: “Several investigations have shown the relationship between preoperative and postoperative function.” This statement is extremely important, which is why it must be cited.

Line 290: “(Numerical et al.)” check if this should actually be presented here

Reviewer 2 ·

Basic reporting

OK

Experimental design

The design is coherent with the research question

Validity of the findings

Some results are likely to change based on the new risk of bias and as I suggest adding the GRADE system, this will likely affect the quality of the recommendation.

Additional comments

Thanks for the opportunnity.

First. The aim of the study “Objective: In this study, we compared the eûects of early and delayed rehabilitation on the function of patients after rotator cuû repair by Meta-analysis to find interventions to promote the recovery of shoulder function.

The aim there are not relationship with the title. Please change the title.

Line 100: They define that only RCTs will be included, and in the exclusion criteria they mention that NO RCTs will be excluded. This information has a contradiction. If they mention RCTs in the inclusion criteria, it is enough to understand that other designs will be left out. Please revise.

Line 109: It is not clear which MESH terms were used, since it only mentions "English terms" in a simple way.

Please change the ROB to ROB 2 analyses.

I can't find the SOF table to review (GRADE)

There is no prior report of the MCID to know if the interventions reached the threshold. Please add.

Why didn't they use some sensitivity analysis for the outcomes? considering the age level etc. or early exercise time. It would be interesting to do it.

---

## Round 0.2 · Major Revisions

· Academic Editor

Major Revisions

While the authors have addressed the vast majority of the comments, the change tracking manuscript is not acceptable in its current form as the authors have apparently copied/pasted from the original manuscript and the entire change tracking document shows only the changes tracking and not the specific changes made. Reviewers need to be able to see and know where specific edits were made. The PeerJ team recommends authors to resubmit the change tracking manuscript showing the original text and the change tracking text. Additionally, the response letter must accurately indicate the line numbers where the modifications were made.

·

Basic reporting

First of all, I appreciate the thoughtfulness of the comments. A notable improvement is seen in writing, the integration of references and the conceptual framework of the research.

Experimental design

With the corrections, the way in which they carried out the investigation is much clearer.

Validity of the findings

The results are interesting for clinical practice. It opens questions to continue investigating the reasons for the findings.

Additional comments

As a final suggestion prior to the final publication, perhaps it would be convenient in the forest plots to change the scale of these graphs. On the x axis some reach 100 and it is not possible to appreciate the distance of the squares, rhombuses and their deviation lines with respect to the vertical line of the graph.

Reviewer 2 ·

Basic reporting

The authors have not made the changes suggested by the reviewer.

Experimental design

no comment

Validity of the findings

no comment

Additional comments

no comment

---

## Round 0.3 · accepted · Accept

· Academic Editor

Accept

The new version of the manuscript has the requested changes. It is suggested that the current version of the manuscript be accepted for publication.